# Assessment of Gastrointestinal Parasites and Productive Parameters on Sheep Fed on a Ration Supplemented with *Guazuma ulmifolia* Leaves in Southern Mexico

**DOI:** 10.3390/ani10091617

**Published:** 2020-09-10

**Authors:** Emelyne Le Bodo, Jean-Luc Hornick, Nassim Moula, Serrano Aracely Zuñiga, Juan Carlos Martínez-Alfaro

**Affiliations:** 1Department of Animal Resources Management and Nutrition Unit, Faculty of Veterinary Medicine, University of Liège, Quartier Vallée 2, Avenue de Cureghem 6, B43a, 4000 Liège, Belgium; emelynelb@gmail.com; 2Department of Animal Resources Management, Faculty of Veterinary Medicine, University of Liège, FARAH Center, Quartier Vallée 2, Avenue de Cureghem 6, B43a, 4000 Liège, Belgium; Nassim.Moula@uliege.be; 3Department of Salubridad e Higiene, Universidad Autónoma Agraria Antonio Narro, Torreón 27054, Mexico; ara7374@hotmail.com; 4Department of Ciencias Médico Veterinarias, Universidad Autónoma Agraria Antonio Narro, Torreón 27054, Mexico; jcmtzalfaro@gmail.com

**Keywords:** gastrointestinal parasites, condensed tannins, animal performance, energy, protein, sheep, *Guazuma ulmifolia*

## Abstract

**Simple Summary:**

Gastrointestinal (GI) parasite infections represent a serious problem in small ruminant production. This issue is currently potentiated by anthelmintic resistance. Thus, the need for more sustainable control alternatives has increased during recent decades. The use of bioactive forages with condensed tannins (CTs) has shown encouraging results. The local Mexican plant *Guazuma ulmifolia* is cited in ethnoveterinary studies and naturally selected by sheep (Pelibuey breed). *G. ulmifolia* contains a certain amount of CTs. Therefore, this study aimed at evaluating the impact of *G. ulmifolia* in sheep diet on animal performance and GI parasites eggs/oocysts excretion. Twenty-two sheep were randomly distributed into two groups: a control group without *G. ulmifolia* and a test group receiving a diet which contained *G. ulmifolia* fresh foliage at 30% of the total diet dry matter. For 30 days, weight, body condition, and GI parasite load were assessed. The results showed no significant anthelmintic and anticoccidial effects (*p* > 0.05) as well as inconsistent impact on live weight during the 30-days experiment. Nevertheless, considering the bromatological analysis of *G. ulmifolia*, its use as a diet supplement to adjust protein and calcium deficits of poor-quality forage is justified.

**Abstract:**

This study aimed at evaluating the impact of a diet supplemented with *Guazuma ulmifolia* leaves on the productive performance and gastrointestinal parasites (GI) eggs/oocysts shedding of sheep (Pelibuey breed). Twenty-two non-lactating ewes were used. They were randomly allocated to two similar groups in terms of age, weight, body condition score (BCS), and GI parasites status. For 30 days, while the control group (G0) was given basic forage hay of *Cynodon nlemfuensis*, the test group (G1) received a blend of *G. ulmifolia* fresh foliage and *C. nlemfuensis* hay (respectively 30 and 70% of the diet on a DM basis). Animals were housed in separate covered pens. Quantitative coprological analysis was performed. Weight and BCS were evaluated. The daily feed amounts given and refused were measured in order to assess nutrient components intake (minerals, energy, and nitrogen). The results showed no significant differences between G1 and G0 regarding the anthelmintic and anticoccidial effects (*p* = 0.57, *p* = 0.91, respectively). Similarly, no significant overall impact on weight and BCS during the experiment (*p* = 0.77, *p* = 0.32, respectively) was observed. Nevertheless, concerning the bromatological analysis of *G. ulmifolia*, its use as a diet supplement to address protein and mineral deficits due to a poor-quality diet is fully justified.

## 1. Introduction

In Mexico, sheep, together with goats and other domestic species, have been raised for five centuries. However, ovine production has increased in recent years, due to the growing demand for meat. Currently, sheep farming contributes significantly to the rural economy of the country, especially in central Mexico for the preparation of traditional dishes. In addition to the high profitability of the species, sheep are prolific ruminants that adapt easily to various environments and make adequate use of resources available in the different regions of the country [1].

In southwestern Mexico, livestock is managed extensively and greatly promoted by government institutions. Thereby, the vegetation was strongly impacted and the original landscape modified, in which corn crop fields were replaced by pastures [2]. It has been documented that trees dispersed in pastures or as a live fence perform multiple functions in production units [2]. For example, tree and shrub foliage with forage potential are considered as an indirect alternative feed in the diet of ruminants, mainly in the time of forage shortage. This alternative aroused some researchers’ interest [3,4,5,6].

In previous studies, the area’s producers were surveyed, allowing for identification of the most locally used tree and shrub species with forage potential [5,6]. In addition to this, studies on sheep ingestive feeding behavior related to tropical trees and their forage potential were conducted. This highlighted the social importance of the *Guazuma ulmifolia* tree (Malvaceae) and revealed that both its foliage [3,5,6] and fruits [4] have a significantly higher preference rate in sheep and are valuable from a nutritional point of view.

Endoparasites are one of the major issues in sheep production. Gastrointestinal parasitic infections have a strong impact on health and production of small ruminants. Thus, the generated losses decrease farm productivity. Without proper internal parasites control, fecundity, milk yield, and carcass value suffer and, in severe cases, death may occur [6,7].

In order to restrain this problem, anthelmintics are widely used. Their use has proven to be an effective control solution. However, within recent years, the prevalence of anthelmintic resistance has risen sharply all around the world including Mexico [8,9,10]. Besides, even if new molecules could emerge, resistance would build up quickly [11] and this issue would remain. Thus, more sustainable alternative control strategies are required [6,12]. The use of bioactive forages that contain tannins is part of the sustainable integrated management strategy. Indeed, it was shown that condensed tannins have antiparasitic effects that include anthelmintic [13,14,15] and anticoccidial activity [16,17]. Besides, other studies revealed that CTs also display nutritional benefits due to protection of dietary protein from excessive fermentation in the rumen. These benefits include improved growth, milk yields, and fertility, and increased resilience to some intestinal parasites [18].

Some studies on *G. ulmifolia* reported CTs concentrations in this tree. Values in foliage varied from 0.23% to 6.5% DM [4,19,20]. However, some investigations reported various phytotherapeutic properties such as anti-diabetic [21,22], hypotensive and vasorelaxant [23,24], antiulcer [25,26], antibacterial [27,28], and antiviral [29] activities. However, only little information on the effects of *G. ulmifolia* on gastrointestinal (GI) parasites is available in the literature, up to now. Only one study investigated the impact of *G. ulmifolia* which was included at 10% of the total diet dry matter over *Haemonchus contortus* in kids [19]. This study, conducted in the state of Guerrero, Mexico, compared *G. ulmifolia* with two other common tropical tree foliage, *Pithecellobium dulce* and *Acacia cochliacantha*. It was concluded that *P. dulce* and *A. cochliacantha*, which present higher CTs concentrations, had potential control over *H. contortus*, contrary to *G. ulmifolia*.

Therefore, the purpose of this study was to evaluate the effects of high-*G. ulmifolia* fodder supply in a period of feed shortage on diet characteristics, animal performance (body condition score, corporal weight), and GI parasites in the sheep of southern Mexico.

## 2. Materials and Methods

### 2.1. Environment, Animal, Plant Material, and Experiment

The experimental procedures used in the current experiments were carried out according to the Official Mexican Rule for technical specifications related to the production, care, and use of laboratory animals (NOM-062-Z00-1992, -003-Z00-1994, -051-Z00-1995, -029-ZOO-1995, -033-ZOO-1995) [30]; the protocol number is PYE-01-Protocolo-2019.

The study was conducted in the Sacrosanto Ranch, Municipality of Jiquipilas, Chiapas, located 16°34′20″ LN and 93°30′50″ LW and 610 m above sea level. According to the Köppen Classification, modified by García [31], the climate is Aw1 (i’) g: warm sub-humid with an annual mean temperature >22 °C, little temperature variation (between 5 and 7 °C), the hottest month before the summer solstice (i.e., June), and rains in summer. The typical vegetation consists of a tropical forest with tree species having potential forage of secondary succession such as *G. ulmifolia.* The presence of the latter is natural in the central region of Chiapas, Mexico. The test was carried out from 9 July to 7 August, 2019, and included only one step, i.e., a 30-day adaptation and measurement.

A total of 22 empty female Pelibuey sheep, 1 to 2 years old, were used. They were randomly allocated to two homogeneous groups in terms of age, body condition (weight, score), and parasitological status as assessed by a McMaster technique [32] (Table 1). The animals were provided with a mineral salt block and water on an ad libitum basis. The control group (G0) was only offered hay of “Estrella of Africa” (*Cynodon nlemfuensis*) as usually practiced in the region. The test group (G1) received a blend of 30% *G. ulmifolia* fresh foliage in addition to 70% *C. nlemfuensis* hay, all together on a DM basis. As the voluntary feed forages intake was not known, a value of 2.1% body weight DM was assumed [33]. Hence, to limit the possible deficiencies in DM requirements, the amount of diet offered was calculated based on the value of 3% body weight DM per day. *C. nlemfuensis*, as hay, was assumed to contain around 95% of DM and *G. ulmifolia*, as fresh foliage, 50% of DM. Each group was offered two meals per day. Body condition score (using the Walkden-Brown scoring) [34] and body weight were measured 3 times (on day 0, day 19, and day 29) and individual coprological analyses were done twice before the start of the trail (day 1 and day 2) and every 5 days thereafter. Each group had a 20 m^2^ area, of which 15 m^2^ was roofed to protect the animals from environmental conditions such as sun and rain. Linear feeders with 11 wood divisions were available in both groups, which allowed each sheep to have their own space for feed consumption. In addition, in the test group (G1), feeder wood divisions were extended along the whole body length, allowing for a complete separation between each sheep. During consumption of *G. ulmifolia* foliage, sheep were immobilized ensuring that each one of them consumed their assigned ration. Feed refusal was measured daily: individually after each meal for *G. ulmifolia*, and by group before offering the morning meal for *C. nlemfuensis*. Thus, the daily feedstuff intakes were calculated through the difference between the amounts offered and not ingested.

### 2.2. Sample Collection and Chemical Analyses

Gastrointestinal parasite fecal eggs and oocysts were counted according to a McMaster technique with a sensitivity of 30 EPG [32,35]. Feces were collected in the morning and conserved individually in a cool place with ice. Coprological analyses of samples (30 EPG detection limit), pooled per group, were performed during the day.

Feed analyses were carried out on three samples of each plant collected at different times of the study (end of July, beginning of August, and end of the experiment).

The chemical composition analyses of the feed samples were carried out according to AOAC procedures [36]. Dry matter (DM) content was determined from a 5-g-test sample maintained at a temperature of 105 °C for 24 h [37]. Crude ash was determined from a 1-g-test sample incinerated in an oven at 550 °C for 4 h. Crude protein (CP) was determined by the Kjeldahl method (N × 6.25); crude fiber (CF) by the method of Weende (Method N° 978.10). Fractions of neutral detergent fiber (NDF) were determined by following the technique proposed by Van Soest et al. [38]. Finally, major macrominerals such as calcium, potassium, sodium, and magnesium were determined by atomic absorption spectrophotometry. Phosphorus was measured by colorimetry. The determination of trace elements—Cu, Fe, Zn, and Mn—was carried out by atomic absorption spectrophotometry.

Tannin concentrations were evaluated using a methylcellulose precipitable tannin assay (MCP) for the total tannins [39], and the ethanolysis method for the CTs.

### 2.3. Calculations and Statistical Analyses

The adequate allocation of animals into groups was verified according to a Student t-test and normality of the variables was assessed. Calculations of forages’ net energy, intestinal digestible protein (IDP), and rumen degradable nitrogen balance (RDNB) were performed according to Institut National de Recherche Agronomique formulas [33].

The substitution rate of *G. ulmifolia* was calculated according to the following equation:(1)$$\mathrm{Substitution}\;\mathrm{rate}=\frac{C.nlemfuensis\;\mathrm{intake}\;\mathrm{GO}-\;C.nlemfuensis\;\mathrm{intake}\;G1\;}{G.\;ulmifolia\;\mathrm{intake}\;}$$

For serial data, a mixed model performed with the SAS statistical package was used [40]. Thus, compound symmetry covariance structures were taken into account. Multiples comparisons were performed according to the Tukey test. Polynomial contrasts were performed in order to assess linear or quadratic effects of time on the parameters.

## 3. Results

### 3.1. Nutrient Values and Diet-Associated Calculations

The forages’ chemical composition is reported in Table 2. Although crude ash levels were similar between the two forages, *G. ulmifolia* presented a CP content higher than 15% DM, i.e., more than twice the content found in *C. nlemfluensis*. Furthermore, levels of different fiber families were lower in *G. ulmifolia* than in *C. nlemfluensis*, especially CF: less than 25% DM vs. close to 35% DM. Therefore, DM net energy value was almost 20% higher in *G. ulmifolia* than in *C. nlemfuensis*. The calculated IDP was almost doubled in *G. ulmifolia* when compared to *C. nlemfuensis* and RDNB shows an excess close to 7% DM in *G. ulmifolia*, while the value is negative in *C.*
*nlemfuensis*. *G. ulmifolia* showed especially large contents of Ca and K (about 19 g/kg DM) and high Cu levels (more than 30 mg/kg DM). Condensed tannins concentration was close to 1% DM for *G. ulmifolia*.

### 3.2. Feed Intake and Animal Performance

The average refusal rate for *G. ulmifolia* was 2.2% of the total weight offered, and it decreased over time, reaching 0.9% at the end of the experiment. By opposition to G1, net energy intake in G0 was slightly lower than the theoretical requirement (Figure 1). This deficit of ingested energy was steeper during the 10 last days of the experiment (average 2254 kJ/day). However, in both groups, IDP intake was higher than the recommended values and largely higher in G1.

In both sheep groups, mineral supplies were adequate and casual deficiency in sodium and zinc was compensated by the mineral salt rock. In G1, with 30% of *G. ulmifolia*, the ratio Ca/P at 3.6 was higher than the recommended values (i.e., 1.1–2.1) [41].

Owing to medium intake of *G. ulmifolia* of 213 g DM/day, sheep from G1 were provided on average 0.075 g CT/kg/day.

Considering the shifts in forage intake after *G. ulmifolia* incorporation in the diet, the substitution rate of *C. nlemfuensis* was estimated at 53%.

The effects of *G.ulmifolia* on animal performance (averages on the entire experiment time) are reported in Table 3. No significant differences were observed between G0 and G1 for all the parameters recorded.

The evolution of animals’ BCS and body weight in the course of the experiment is reported in Figure 2 and Figure 3. There were significant effects of time (*p* < 0.01) and the interaction of treatment × time (*p* < 0.05) on the first parameter. On day 19, the BCS of the animals from G1 was significantly higher when compared to G0. On day 29, this difference lost its significance, although it kept the same sign numerically.

No significant time effect over weight in general nor differences between the weights of the groups over time were observed, although a trend was noted for the time impact (*p* < 0.1). A gap between the groups seemed to emerge over time. Indeed, whereas G0 followed a downward slope, the test group appeared to be more stable (Figure 3). 

### 3.3. Quantitative Parasitological Analysis

Coprological analyses allowed to identify three main GI parasites: Gastrointestinal strongyle-“type” eggs: term which includes GI strongyle eggs with morphological similarities, such as, for the most common genus, Haemonchus, Ostertagia, Trichostrongylus, and Cooperia [6];Strongyloides papillosus eggs;*Eimeria* spp. oocysts. 

The effects of *G.ulmifolia* on parasite counts (averages on the entire experiment time) are reported in Table 3. No significant differences were observed between G0 and G1. 

From a dynamic point of view, the number of GI strongyle-“type” EPG showed no significant difference between G0 and G1 at any time (Figure 4). However, the evolution with time revealed a very highly significant linear increase (*p* < 0.001). The number of *Eimeria* spp. OPG also showed no significant difference between G0 and G1, at any time. However, a significant linear decrease with time (*p* < 0.05) was observed (Figure 5). Finally, the eggs counts of *S. papillosus* (Figure 6) showed no significant difference between G0 and G1 at any time.

## 4. Discussion

### 4.1. Nutrient Values and Diet-Associated Calculations

Considering the chemical composition of the two feed resources, *G. ulmifolia* foliage could be considered as a high-quality forage, with energy levels close to that reported in concentrates and valuable protein levels. Additionally, the rate of substitution concerning hay of *C. nlemfuensis*, and the great appetence for *G. ulmifolia*, associated with very low refusal rates, are in agreement with the “concentrate-like” effects of this forage. *C. nlemfuensis* is considered presently as a medium–poor-quality forage. Noteworthy, *G. ulmifolia* is a very good source of calcium, with levels higher than those generally observed in leguminous plants. It should be kept in mind that calcium excess could interact with zinc absorption and, to a lesser extent, copper [41,42].

*G. ulmifolia* foliage supplementation in a poor diet allows meeting metabolizable energy (ME) and digestible protein (MP) requirements for maintenance while sparing forage during the shortage season. Assessment of the more adequate percentage of *G. ulmifolia* foliage in order to establish energy, IDP, and RDNB equilibrium found it to be 16% of the diet DM. So, in comparison with the present study, sparing use of *G. ulmifolia* is possible while preserving maintenance performance. Moreover, *G. ulmifolia*, by contributing to provide a more equilibrated diet at various levels, helps to avoid nutritional deficiencies. This is an important consideration because not only does an appropriate nutritional supply (in terms of degradable energy and protein or minerals) preserve the ruminal flora quality [42,43], it also improves small ruminant resistance against parasites. Thus, sheep in an undernourishment state (protein in particular) are less resilient and resistant to GI parasites [6]. In turn, subclinical parasitism amplifies nutrient requirements. Indeed, in general, due to the greater use of amino acids to mount the immune response and the increased GI tract tissue protein turnover, internal parasitism has an impact on the MP requirement, which seems to be relatively greater than the ME requirement [44,45]. Consequently, the optimum ME/DP dietary ratio is influenced. 

In this way, it has been shown that CTs, by binding to dietary protein and reducing rumen proteolysis, increased the proportion of dietary protein in the small intestine and consequently protein absorption [46]. Their benefits are more likely when dietary protein exceeds the requirements, which was the case in this experiment (Figure 1). However, this CTs capacity seemed to be correlated with both the structure of proteins and CTs themselves. Indeed, in recent studies summarized in Mueller-Harvey’s review [18], the mean degree of polymerization length (mDP) of CT (which might be associated with the specific CT type in certain cases) seemed to be correlated with the bioactive effect on proteins. Thus, the only proof of the presence of condensed tannins, even in important proportions, is not sufficient to warrant benefits due to a certain type of CTs.

### 4.2. Feed Intake, Animal Performance, and Gastrointestinal Parasites

Supplementation with *G. ulmifolia* showed no significant effect on weight gain. The lack of an adaptation period to the plant, the management of the sheep, and the short evaluation time could partly explain this. Indeed, alongside the experiment time, weight differences between G0 and G1 increased numerically and the percentage of *G. ulmifolia* refusal decreased (from 7.2% for the first five days to 0.9% for the five last days). An evaluation with a larger study time and an adaptation period might prove the impact of *G. ulmifolia* on weight, as in the article of Rubio et al. [3]. Moreover, as a consequence of the nutritional issues above-mentioned, GI nematode infection could have a significant impact on the metabolic status of sheep, including weight and BCS. This clinical impact increased alongside parasitic load. In this experiment, the GI strongyle egg shedding increased during the experiment, reaching a level (>1000 EPG) which is considered to be associated with heavy parasite burden and significant effects on health and productivity in sheep [6]. This gradual increase in egg shedding after 15 days of trial could be partly explained by the climate that impacts helminth development. Thus, an infestation could have occurred on the pasture at the start of the rainy season (i.e., from June) before sheep batching. The prepatent period varied between three and five weeks according to the helminth species [47], and a significant consecutive excretion of eggs in feces could have gradually taken place from the second part of the experiment (i.e., mid-July). This increase in both groups could also be explained by the stress induced by allotment in a limited space. Besides, some one-year-old animals, which face greater parasite infections, were part of the groups. In older animals, repeated parasitic exposures allow for the establishment of immunity and thus parasitic resistance. Indeed, after a full year of grazing, most sheep develop a high degree of immunity which regulates the GI helminth number [6,12,48]. Hence, it would have been more appropriate to perform the fecal egg counts individually. This phenomenon should be taken into consideration when comparing results between different literature sources addressing effects of CT on intestinal parasites.

The evaluation of nutritional supplies and intakes such as energy, protein, and minerals suggested no severe deficiency likely to favor parasitological load. However, the high proportion of *G. ulmifolia* in the diet of G1 in comparison to the study of León-Castro et al. [19] (30% versus 10% DM of the total diet, respectively) led us to expect that a possible effect would emerge. Unfortunately, as well as the previous study, the present supplementation with *G. ulmifolia* showed no significant antiparasitic effect. When comparing the chemical compositions of *G. ulmifolia* foliage in both experiments, the concentration of CT was higher in León-Castro et al.’s experiment (4.1% DM versus 1.1% DM of this study) [19]. This difference highlighted the fact, described in the review of Mueller-Harvey et al. [18] that CT concentration varies greatly not only between plant species but between accessions as well. Yet, the variation of CT according to the analysis technique is also noticeable [49]. Nevertheless, the parasitological impact of CT in diet fluctuates according to the CT type (structural composition, ratio procyanidin-type/prodelphinidin-type, mDP) too [18]. Thus, it is not excluded that the lack of significant antiparasitic impact could have been due to an insufficient concentration or to an inadequate structural type of the CTs contained in *G. ulmifolia* foliage. In addition, the short treatment period probably played a role in this absence of efficiency. Indeed, Cenci et al. [50] did not observe significant effects of a diet containing 3.2 g CT on lamb growth or inconsistent effects of this treatment on various helminth species, after a treatment lasting for 84 day. In our experiment, no effects on sheep were observed with estimated amounts of CT intake close to 2.3 g/day, offered during a lower period. Iqbal et al. [51] also observed inconsistent anthelmintic effects of CT on *H. contortus* in lambs fed 2 to 3% CT in their diet, with the level of 3% appearing effective only after 60d of the experiment. When considering the effects of CT on coccidiosis, Saratsis et al. [17] reported high variability of results in ewes and lambs offered CT-rich feeds (sainfoin and/or carob pods) for six–eight weeks. Finally, it must be pointed out that CTs are possibly less or not effective on abomasal worm species, as observed by Athanasiadou et al. [52] with a tannin-rich tree extract (Quebracho), offered to sheep during 3 day at 8% *w/w*. In our experiment, sheep examined were positive for gastrointestinal strongyle eggs at fecal parasitological analysis. However, the identification of these nematodes at the genus/species level is not possible on the basis of the morphological features of their eggs [7].

These considerations suggest that higher amounts of *G. ulmifolia* could be effective to control certain sheep gastrointestinal parasites in southern Mexico but raise the question of potential over-harvesting of the species and nutritional constraints for diet balancing.

## 5. Conclusions

*G. ulmifolia* has proven to be a valuable energy and nutrient source. Its use as a diet complement to adjust protein deficit in a basic–medium ration is valuable, considering that adequate nutrient intake is very important for sheep productivity and resistance to parasitological infection. However, in the limited allocation time of this study, the proposed *G. ulmifolia* supplementation had a limited impact on animal weight and BCS.

Despite the antiparasitic and nutritional benefits reported for CT in the literature, no apparent effect was highlighted for CT contained in *G. ulmifolia* foliage tested in this study. A longer lasting experiment with a higher percentage of this plant’s leaves is required to confirm these observations. Moreover, structural composition analysis of CTs proper to *G. ulmifolia* would be interesting in order to investigate whether the lack of effect could possibly be ascribed to an insufficient concentration of CTs or to their type.

## Figures and Tables

**Figure 1 animals-10-01617-f001:**
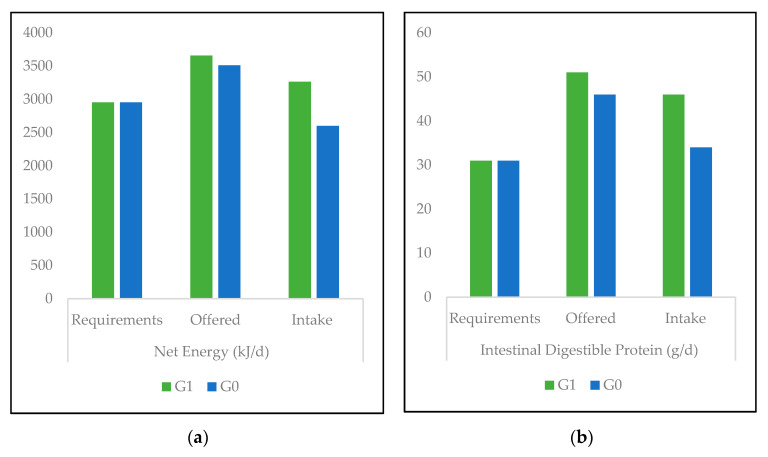
Energy (**a**) and protein (**b**) requirement, offered and ingested in sheep fed on *C. nlemfuensis* hay alone (G0) or *G. ulmifolia* foliage and *C. nlemfuensis* hay (G1). Calculations according to the INRA system. kJ: kilojoule, d: day, g: gram.

**Figure 2 animals-10-01617-f002:**
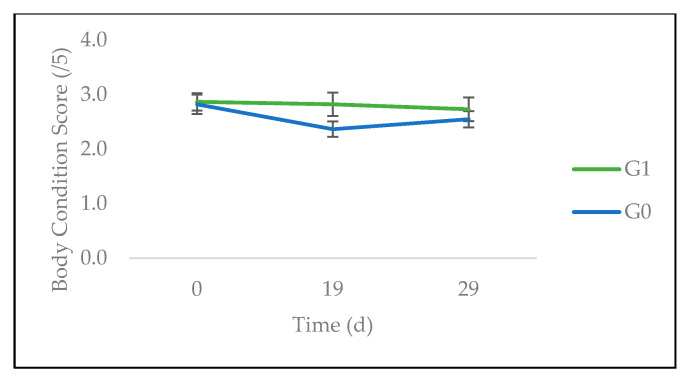
Evolution of the body condition score of sheep fed on *C. nlemfuensis* hay alone (G0) or *G. ulmifolia* foliage and *C. nlemfuensis* hay (G1). d: day.

**Figure 3 animals-10-01617-f003:**
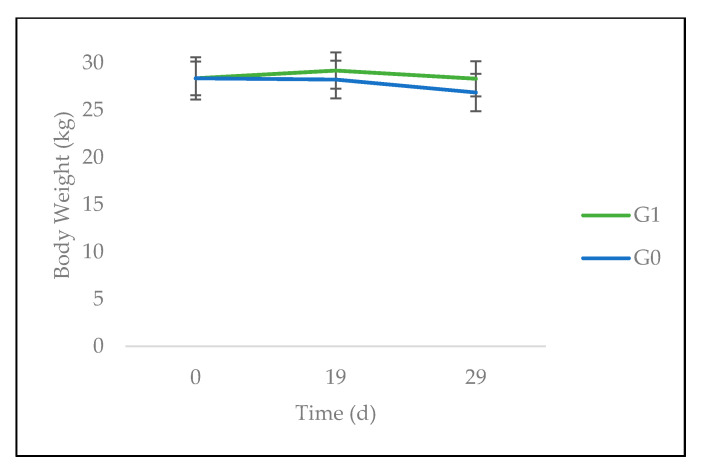
Evolution of body weight of sheep fed on *C. nlemfuensis* hay alone (G0) or *G. ulmifolia* foliage and *C. nlemfuensis* hay (G1). d: day.

**Figure 4 animals-10-01617-f004:**
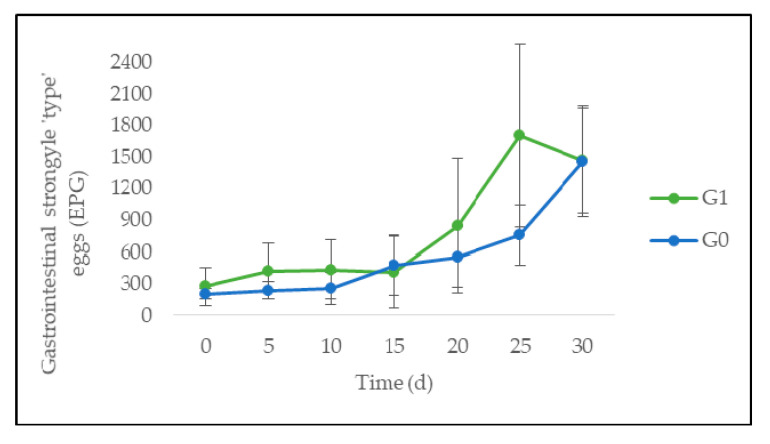
Evolution of gastrointestinal strongyle-“type” eggs per gram of feces (EPG) of sheep fed on *C. nlemfuensis* hay alone (G0) or *G. ulmifolia* foliage and *C. nlemfuensis* hay (G1). d: day.

**Figure 5 animals-10-01617-f005:**
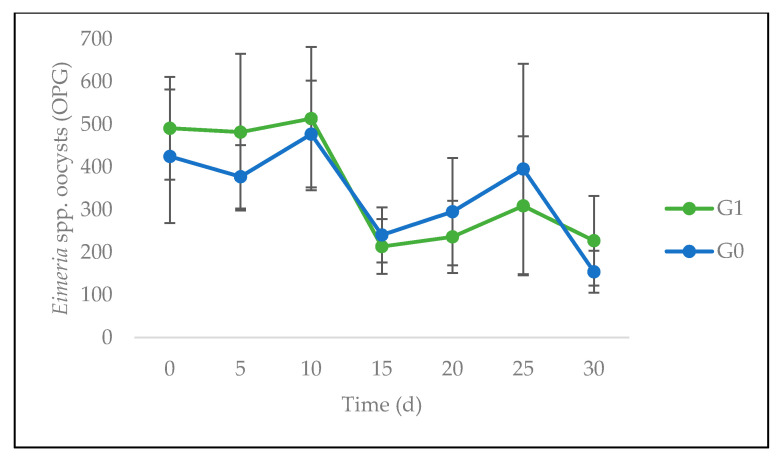
Evolution of *Eimeria* spp. oocysts per gram of feces (OPG) of sheep fed on *C. nlemfuensis* hay alone (G0) or *G. ulmifolia* foliage and *C. nlemfuensis* hay (G1) spp. d: day.

**Figure 6 animals-10-01617-f006:**
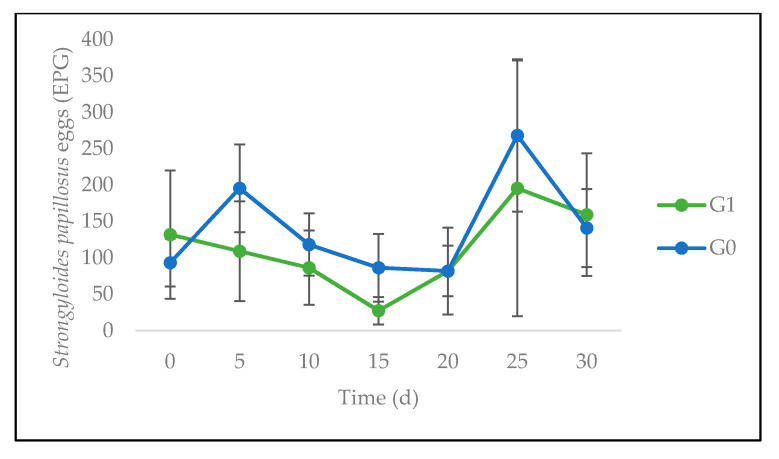
Evolution of *Strongyloides papillosus* eggs per gram of feces (EPG) of sheep fed on *C. nlemfuensis* hay alone (G0) or *G. ulmifolia* foliage and *C. nlemfuensis* hay (G1). d: day.

**Table 1 animals-10-01617-t001:** Group characteristics according to initial animal parameters (mean ± SE).

Parameters	Group		*p*-Value
	G0	G1	
Age (year)	1.6 ± 0.6	1.6 ± 0.6	1
Body weight (kg)	28.4 ± 0.5	28.4 ± 1.1	1
BCS (/5)	2.8 ± 0.4	2.9 ± 0.4	0.8
Gastrointestinal strongyle-“type” eggs (EPG)	202 ± 10	277 ± 35	0.7
*Eimeria* spp. oocysts (OPG)	430 ± 24	493 ± 17	0.7
*Strongyloides papillosus* eggs (EPG)	93 ± 11	132 ± 25	0.7

SE: standard error; G0: sheep offered only *C. nlemfuensis* hay; G1: sheep offered 30% DM *G. ulmifolia* foliage and 70% DM *C. nlemfuensis* hay; BCS: body condition score; EPG: eggs per gram; OPG: oocysts per gram.

**Table 2 animals-10-01617-t002:** Chemical composition, energy, and protein values of *G. ulmifolia* (fresh foliage) and *C. nlemfuensis* (hay) offered to sheep.

Parameters	*G. ulmifolia (foliage)*		*C. nlemfuensis (hay)*	
	Average	SD	Average	SD
Dry matter (% Fresh matter)	38.7	0.8	83.0	0.9
Crude ash (% DM)	10.7	1.3	10.5	2.7
Crude protein (% DM)	15.4	2.3	6.9	2.3
Crude fiber (% DM)	22.9	2.4	34.5	4.6
NDF (% DM)	62.7	3.8	68.5	8.2
Total tannins (% DM)	0.79	0.05	/	
Condensed tannins (% DM)	1.05	0.04	/	
Calcium (g/kg DM)	19.0	5.0	4.2	0.5
Phosphorus (g/kg DM)	3.1	0.4	1.7	0.9
Potassium (g/kg DM)	18.6	2.3	14.9	0.5
Sodium (g/kg DM)	0.1	0.1	1.1	1.1
Magnesium (g/kg DM)	4.4	0.7	1.7	0.3
Copper (mg/kg DM)	30.2	18.5	7.3	5.0
Iron (mg/kg DM)	86.5	23.8	368.5	261.9
Manganese (mg/kg DM)	49.9	5.5	79.4	8.4
Zinc (mg/kg DM)	31.0	14.4	32.7	6.4
Net energy (kJ/kg DM)	5693	72	4726	106
IDP (% DM)	9.0	0.5	6.2	0.8
RDNB (g)	66.5	21.9	−9.8	7.9

SD: standard deviation; DM: dry matter; NDF: neutral detergent fiber; IDP: intestinal digestible protein (according to INRA system); RDNB: rumen degradable nitrogen balance.

**Table 3 animals-10-01617-t003:** Mean weight, BCS, and parasitic loads of sheep offered hay of *C. nlmenfuensis* alone (G0) or supplemented with fresh foliage of *G. ulmifolia* (G1) (mean ± SE).

Parameters/Parasites	Group		*p*-Value
	G0	G1	
Body weight (kg)	27.8 ± 2.1	28.6 ± 1.9	0.77
BCS (/5)	2.6 ± 0.2	2.8± 0.2	0.32
Gastrointestinal strongyle-“type” eggs (EPG)	562 ± 241	791 ± 448	0.57
*Eimeria* spp. oocysts (OPG)	338 ± 123	353 ± 130	0.91
*Strongyloides papillosus* eggs (EPG)	141 ± 80	113 ± 55	0.75

SE: standard error; G0: sheep offered only *C. nlemfuensis* hay; G1: sheep offered 30% DM *G. ulmifolia* foliage and 70% DM *C. nlemfuensis* hay; BCS: body condition score; EPG: eggs per gram feces; OPG: oocysts per gram feces.

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
