# Peer review of "Assessment of Gastrointestinal Parasites and Productive Parameters on Sheep Fed on a Ration Supplemented with Guazuma ulmifolia Leaves in Southern Mexico"

_animals, 2020, doi:10.3390/ani10091617_

Round 1
Reviewer 1 Report
Given the context of increasing interest in sheep production in Mexico and the ever present challenge of parasitic infestation this is a useful piece of work to show a lack of anti-parasitic effects of G. ulmifolia. The experiment seems to have been well designed and conducted with approriate variables measured. generally the paper is well written although in places the English language needs some attention, but generally it is of a good standard.
My only question is whether the experimental period i.e. 30 days was sufficiently long to show effects. The authors do conclude that longer term studies are needed.
Author Response
Dear Reviewer,
Thank you for your comments.
Hereafter you will find the answers and corrections made following your observations:
|
334- |
My only question is whether the experimental period i.e. 30 days was sufficiently long to show effects. The authors do conclude that longer term studies are needed. Indeed, a longer experiment would have be more advisable. More considerations are given in relation with the short experimental period in the discussion. |
English editing has been addressed with help for this manuscript. In the same way, topographical errors were corrected in harmony with 'Instructions for Authors' on the journal website.
All those corrections are clearly highlighted with the "Track Changes" function in Microsoft Word.
Best regards,
The authors.
Reviewer 2 Report
This is a very interesting and really well-written manuscript concerning the evaluation of parasitic and productive parameters in sheep fed a blend of fresh foliage of the plant Guazuma ulmifolia. Nevertheless, all parasitological terms are incorrectly reported. Moreover, and most important, probably the plant did not have any effects against parasites due to its low content of condensed tannins or to a short treatment period (see Cenci et al. Effects of condensed tannin from Acacia mearnsii on sheep infected naturally with gastrointestinal helminthes Vet. Parasitol., 144 (2007), pp. 132-137; Morais-Costa et al. In vitro and in vivo action of Piptadenia viridiflora (Kunth) Benth against Haemonchus contortus in sheep. Vet Parasitol. 2016;223:43-49. doi:10.1016/j.vetpar.2016.04.002; Iqbal et al. Direct and indirect anthelmintic effects of condensed tannins in sheep. Vet Parasitol. 2007;144(1-2):125-131. doi:10.1016/j.vetpar.2006.09.035; Saratsis et al. The effect of sainfoin (Onobrychis viciifolia) and carob pods (Ceratonia siliqua) feeding regimes on the control of lamb coccidiosis. Parasitol Res. 2016;115(6):2233-2242. doi:10.1007/s00436-016-4966-9) or because the sheep examined were infected mainly by gastrointestinal strongyle species localised in the abomasum on which condensed tannins are less effective (Athanasiadou et al., Direct anthelmintic effects of condensed tannins towards different gastrointestinal nematodes of sheep: in vitro and in vivo studies. Vet. Parasitol., 99 (2001), pp. 205-219). These important topics should be adequately discussed in the manuscript. For all these reasons, it is advisable that the revised manuscript is reviewed by a veterinary parasitologist experienced in sheep gastrointestinal parasites.
More details are given below.
In tables 1 and 3 and throughout the manuscript: the names of parasite genera should be written in italics. Moreover, it is better to replace “Trichostrongylus type eggs” with “gastrointestinal strongyles eggs”, “Strongyloides spp. larvated eggs” with “Strongyloides papillosus eggs”, “type of helminths” with “helminth species”, and “MacMaster’s cell count technique” with “a McMaster technique” and add the sensitivity of the McMaster technique used in the study (in EPG). Moreover, it would be advisable to explain better that the resistance against gastrointestinal strongyles of adult sheep consists of a parasite load generally much lower than in first-year grazing sheep, unless they are pregnant.
Author Response
Dear Reviewer,
Thank you for your encouraging comments.
Hereafter you will find the answers and corrections made following your observations:
|
Table 1 (143), Table 3 (217), 238, 246, 253, Figure 4 (256), Figure 6 (260) |
Replace “Strongyloides spp. larvated eggs” with “Strongyloides papillosus eggs”. OK because to date, most veterinary parasitologists consider that all Strongyloides of farm ruminants to belong to the same species, S. papillosus. A heterogeneity was found for cattle recently but not for sheep. (Thamsborg et al., 2017). Replace “Trichostrongylus type eggs” with “gastrointestinal strongyles eggs”, In this case, with the McMaster technique we found eggs typically classified as “Trichostrongylus-type” or “HOTC-type” due to their morphological similarities. This term include several gastrointestinal nematode genera [Haemonchus, Ostertagia, Trichostrongylus, and Cooperia (HOTC) for the most common, but also Teladorsagia, Spiculopteragia, Apteragia, Bunostomum, Chabertia, and Oesophagostomumi]. However, due to distinct morphological differences, it did not include all of them (Nematodirus, Marshallagia, Aonchotheca, Strongyloides, Gongylonema, Skrjabinema, and Trichuris). (Pugh et al., 2021) Indeed, for us, it is not correct to write “gastrointestinal strongyle eggs”. Nevertheless, in order to adjust as demand while being more specific, we changed for “gastrointestinal strongyle ‘type’ eggs” (with a quick explanation in the beginning of the Results). The names of parasite genera should be written in italics. OK |
|
311 |
Replace “type of helminths” with “helminth species” OK |
|
121, 149-151 |
Replace “MacMaster’s cell count technique” with “a McMaster technique” OK and add the sensitivity of the McMaster technique used in the study (in EPG). We added “The mean sensitivity of this method is 47.3% for 50 EPG and 100% for 500 EPG. These results, described by in the study of Becker and collaborators [33], were found using common GI helminth parasites.” |
|
334-347 |
Moreover, and most important, probably the plant did not have any effects against parasites due to its low content of condensed tannins or to a short treatment period (see Cenci et al. Effects of condensed tannin from Acacia mearnsii on sheep infected naturally with gastrointestinal helminthes Vet. Parasitol., 144 (2007), pp. 132-137; Morais-Costa et al. In vitro and in vivo action of Piptadenia viridiflora (Kunth) Benth against Haemonchus contortus in sheep. Vet Parasitol. 2016;223:43-49. doi:10.1016/j.vetpar.2016.04.002; Iqbal et al. Direct and indirect anthelmintic effects of condensed tannins in sheep. Vet Parasitol. 2007;144(1-2):125-131. doi:10.1016/j.vetpar.2006.09.035; Saratsis et al. The effect of sainfoin (Onobrychis viciifolia) and carob pods (Ceratonia siliqua) feeding regimes on the control of lamb coccidiosis. Parasitol Res. 2016;115(6):2233-2242. doi:10.1007/s00436-016-4966-9) or because the sheep examined were infected mainly by gastrointestinal strongyle species localised in the abomasum on which condensed tannins are less effective (Athanasiadou et al., Direct anthelmintic effects of condensed tannins towards different gastrointestinal nematodes of sheep: in vitro and in vivo studies. Vet. Parasitol., 99 (2001), pp. 205-219). These important topics should be adequately discussed in the manuscript. For all these reasons, it is advisable that the revised manuscript is reviewed by a veterinary parasitologist experienced in sheep gastrointestinal parasites. We have adjusted the end of the discussion in relation with your proposals. |
|
316-319 |
It would be advisable to explain better that the resistance against gastrointestinal strongyles of adult sheep consists of a parasite load generally much lower than in first-year grazing sheep, unless they are pregnant. More explanations were given. |
|
|
|
Moreover, English editing has been addressed with help for this manuscript. In the same way, topographical errors were corrected in harmony with 'Instructions for Authors' on the journal website.
All those corrections are clearly highlighted with the "Track Changes" function in Microsoft Word.
Best regards,
The authors.
References :
Thamsborg, S.M.; Ketzis, J.; Horii, Y.; Matthews, J. B. Strongyloides Spp. Infections of Veterinary Importance. Parasitology 2017, 144 (3), 280.
Pugh, D.G.; Baird, A.N.; Edmondson M.A.; Passler, T. Internal parasites of sheep, goats, and cervids. In Sheep, Goat, and Cervid Medicine, 3rd ed.; Elsevier: St. Louis, MO, USA, 2021, pp 276–287.
Becker, A.C.; Kraemer, A.; Epe, C.; Strube, C. Sensitivity and efficiency of selected coproscopical methods—sedimentation, combined zinc sulfate sedimentation-flotation, and McMaster method. Parasitol. Res. 2016, 115(7), 2581–2587, doi:10.1007/s00436-016-5003-8.
Reviewer 3 Report
This study describes the addition of Guazuma ulmifolia leaves as a possible feed for the control of gastrointestinal parasites in sheep. In general, this study was well carried out. However, the authors need to improve some details which are found in their manuscript.
The comments and suggestions are added into the manuscript

Author Response
Dear Reviewer,
Thank you for your encouraging comments.
Hereafter you will find the answers and corrections made following your observations:
|
4 |
Change “leaves of Guazuma ulmifolia” to “Guazuma ulmifolia leaves” OK |
|
38 |
Change ‘placed’ to ‘housed OK |
|
38 |
“For thirty days, weight, body condition, and GI parasite load assessed by coprological analyses were obtained.” What are the authors trying to say in this section? I undestand, please rephrase this sentence. We have rephrased it and adjusted the abstract in order not to excess the 200 words demanded. |
|
94 |
Correct “Guerrera” to “Guerrero” OK |
|
147 |
Correct “Analyses” to “Analysis” In this sentence “AnalysIS” (singular) is plural, so analySES is correct (UK dictionary), if you’d rather in US version we could change for analyZES. However we had made a mistake ligne 156 “2.3. Calculations and Statistical AnalysIS” corrected with “2.3. Calculations and Statistical AnalysES”. |
|
271 |
copper [40,Error! Reference source not found.]. Link 42 refreshed (if it was the problem?). |
|
280-282 |
“Conversely, balanced feeding improves resistance to parasites.” I don´t understand this sentence. You say that the animals that consume a diet rich in nutrients the parasites are able to survive in the host, and this is not correct. Please, rephrase this sentence. We have rephrased the paragraph in order to be more correct. |
|
352 |
Change “G. ulmifolia” to “Guazuma ulmifolia” In the Guidelines for authors (https://www.mdpi.com/authors/english-editing), it is mentioned: “Define abbreviations the first time they are mentioned in the abstract, text; also the first time they are mentioned in a table or figure.” It is not mentioned that abbreviation should be written in full at the sentence beginning, if this was what you wanted to note? |
Moreover, English editing has been addressed with help for this manuscript. In the same way, topographical errors were corrected in harmony with 'Instructions for Authors' on the journal website.
All those corrections are clearly highlighted with the "Track Changes" function in Microsoft Word.
Best regards,
The authors.
Round 2
Reviewer 2 Report
mainly due to the lack of a parasitolog
The revised manuscript has improved. However, mainly due to the lack of a parasitologist among the authors of this manuscript, the changes and corrections reported below are required.
Title
Parasites are not parameters but living organisms!
Please, replace the title with: Assessment of Gastrointestinal Parasites and Productive Parameters on Sheep Fed on a Ration Supplemented with Guazuma ulmifolia Leaves in Southern Mexico.
Abstract:
Line 30: replace “parasitological loads” with “gastrointestinal parasites”
Lines 35-36: The load of parasites is evaluated mainly by counting the number of parasites in the gastrointestinal tract. The number of nematode eggs or of coccidian oocysts give only an estimation of the load of these parasites. Therefore, replace “The GI parasites’ load was estimated by coprological analyses” with “Quantitative coprological analysis was performed for the evaluation of gastrointestinal parasites”.
Keywords:
Replace “gastrointestinal parasitism” with “gastrointestinal parasites”
Introduction
Lines 66-67: Helminths and coccidia cause infections. Replace “One of the major issues in sheep production, as a meat source, is the species sensitivity to endoparasitism. Gastrointestinal parasitic infestations have” with “Endoparasites are one of the major issues in sheep production. Gastrointestinal parasitic infections have”
Materials and Methods
Lines 135-139: please replace “Gastrointestinal parasites were counted according to the McMaster technique [32]. The mean sensitivity of this method is 47.3% for 50 EPG and 100% for 500 EPG. These results, described by in the study of Becker et al. [35], were found using common GI helminth parasites.” with “Gastrointestinal parasites were counted according to a McMaster technique with a sensitivity of 50 EPG [32,35].”
Results
Table 3: Replace “Parameters” with “Parameters/Parasites”, “Gastrointestinal strongyle ‘type’ eggs (EPG)” with “Gastrointestinal strongyle eggs (EPG)”. Moreover, spp. should not be written in italics: please, correct.
Line 230: replace “Parasites Load Response to Treatment” with “Quantitative Parasitological Analysis”
Lines 232-236: please replace
“1. Gastrointestinal strongyle ‘type’ eggs: term which include GI strongyles with morphological 232 similarites, such as, for the most common genus, Haemonchus, Ostertagia, Trichostrongylus, and Cooperia [6].
- Strongyloides papillosus eggs
- Eimeria spp. ookysts”
with
“1. Gastrointestinal strongyle eggs
- Strongyloides papillosus eggs
- Eimeria spp. oocysts”
Line 237: delete this sentence
Line 240: please replace “gastrointestinal strongyle ‘type’ eggs load showed” with “the number of gastrointestinal strongyle EPG showed”
Line 242: please replace “Eimeria spp. oocysts load also showed” with “the number of Eimeria spp. OPG also showed”
Line 244: please replace “Finally, the load of Strongyloides papillosus eggs” with “Finally, the number of S. papillosus EPG”
Line 247: please replace “Figure 4. Evolution of parasitic load of gastrointestinal strongyle ‘type’ eggs (EPG) of sheep fed” with “Figure 4. Gastrointestinal strongyle eggs per gram of faeces (EPG) of sheep fed”
Lines 250-251: please replace “Figure 5. Evolution of parasitic load of Eimeria spp. oocysts (OPG) of sheep fed on C. nlemfuensis hay alone (G0) or G. ulmifolia foliage and C. nlemfuensis hay (G1).” with “Figure 5. Eimeria spp. oocysts per gram of faeces (OPG) of sheep fed on C. nlemfuensis hay alone (G0) or G. ulmifolia foliage and C. nlemfuensis hay (G1).”
Lines 253-254: please replace “Figure 6. Parasitic load of Strongyloides papillosus eggs (EPG) of sheep fed on C. nlemfuensis hay alone (G0) or G. ulmifolia foliage and C. nlemfuensis hay (G1).” with “Figure 6. Strongyloides papillosus eggs per gram of faeces (EPG) of sheep fed on C. nlemfuensis hay alone (G0) or G. ulmifolia foliage and C. nlemfuensis hay (G1).”
Discussion
Line 288: please replace “4.2. Feed Intake, Animal Performance, and Parasitic Load” with “4.2. Feed Intake, Animal Performance, and Gastrointestinal Parasites”
Line 297: please replace “parasitic load of nematodes” with “the EPG number of gastrointestinal strongyle eggs”
Line 298: replace “reaching significant parasitism at >1000 EPG” with “reaching a number >1000 EPG, indicating a possible high gastrointestinal strongyle load”
Line 312: please replace “parasitological load” with “parasite infections”
Line 308: please replace “worm number” with “nematode (or helminth) number”
Line 317: please replace “presently” with “of this study”
Line 327: please replace “various worms” with “various nematode (or helminth) species
Lines 334-336: please delete the sentence “In our experiment, sheep examined were infected principally by gastrointestinal strongyle ‘type’ species. This terminology, used with the McMaster technic results, include not only species localized in the abomasum, but mostly [7].” with “In our experiment, sheep examined were positive for gastrointestinal strongyle eggs at faecal parasitological analysis. However, the identification of these nematodes at the genus/species level is not possible on the basis of morphological features of their eggs [7].”
Line 339: please replace “cattle (???)” with “sheep” and “worms” with “parasites”
Lines 347-348: please replace “Considering the antiparasitological and nutritional benefits of CT present in G. ulmifolia foliage no apparent effect was highlighted.” with “Despite the antiparasitic and nutritional benefits reported for CT [insert references], no apparent effect was highlighted for CT contained in G. ulmifolia foliage tested in this study.”
References
Reference n. 6 should be replaced with a more appropriate one (published by parasitologists in an international journal)
References n. 17 and 53 are the same
In the reference n. 51 “Acacia Mearnsii” should be written in italics and the first letter of the name of the species should be in lowercase.
Author Response
Dear reviewer,
Thank you for your comments. In this following letter you’ll find our responses.
“The revised manuscript has improved. However, mainly due to the lack of a parasitologist among the authors of this manuscript, the changes and corrections reported below are required.”
Between the sending of the second manuscript and the receipt of your second opinion we finally managed to have our manuscript proofread by a parasitologist, (Prof. B. Losson, Liège University, Belgium). Thus, we also took into account his corrections / remarks for this second manuscript and the following answers.
Title
Parasites are not parameters but living organisms!
Please, replace the title with: Assessment of Gastrointestinal Parasites and Productive Parameters on Sheep Fed on a Ration Supplemented with Guazuma ulmifolia Leaves in Southern Mexico.
Done
Abstract:
Line 30: replace “parasitological loads” with “gastrointestinal parasites”
Done, Line 32 we completed the term with “eggs/oocysts shedding”.
Lines 35-36: The load of parasites is evaluated mainly by counting the number of parasites in the gastrointestinal tract. The number of nematode eggs or of coccidian oocysts give only an estimation of the load of these parasites. Therefore, replace “The GI parasites’ load was estimated by coprological analyses” with “Quantitative coprological analysis was performed for the evaluation of gastrointestinal parasites”.
Done, your comment regarding the relationship between FEC and parasite load is in relation with the proofreading of Pr Losson. Thus, we modified it all along the text.
Line 38 “Quantitative coprological analyses was performed”, here we have shorten this sentence in order to be in the right number of words for the abstract (200).
Keywords:
Replace “gastrointestinal parasitism” with “gastrointestinal parasites”
Done
Introduction
Lines 66-67: Helminths and coccidia cause infections. Replace “One of the major issues in sheep production, as a meat source, is the species sensitivity to endoparasitism. Gastrointestinal parasitic infestations have” with “Endoparasites are one of the major issues in sheep production. Gastrointestinal parasitic infections have”
Done
Materials and Methods
Lines 135-139: please replace “Gastrointestinal parasites were counted according to the McMaster technique [32]. The mean sensitivity of this method is 47.3% for 50 EPG and 100% for 500 EPG. These results, described by in the study of Becker et al. [35], were found using common GI helminth parasites.” with “Gastrointestinal parasites were counted according to a McMaster technique with a sensitivity of 50 EPG [32,35].”
Done Line 136-139, in the reference it was exactly mentioned a mean sensitivity of 25.9% for 30EPG, so, we took this first value instead of 50EPG.
Results
Table 3: Replace “Parameters” with “Parameters/Parasites”, “Gastrointestinal strongyle ‘type’ eggs (EPG)” with “Gastrointestinal strongyle eggs (EPG)”. Moreover, spp. should not be written in italics: please, correct.
Done for the spp. in italics.
However we explained you in the previous cover letter (undermentioned) why we had chosen the term “Gastrointestinal strongyle ‘type’ eggs (EPG)” instead of “Gastrointestinal strongyle eggs”. Could you explain more why you don’t accept this term?
With this appellation it includes Strongyloides papillosus which isn’t correct (not the same morphology).
“In this case, with the McMaster technique we found eggs typically classified as “Trichostrongylus-type” or “HOTC-type” due to their morphological similarities. This term include several gastrointestinal nematode genera [Haemonchus, Ostertagia, Trichostrongylus, and Cooperia (HOTC) for the most common, but also Teladorsagia, Spiculopteragia, Apteragia, Bunostomum, Chabertia, and Oesophagostomumi]. However, due to distinct morphological differences, it did not include all of them (Nematodirus, Marshallagia, Aonchotheca, Strongyloides, Gongylonema, Skrjabinema, and Trichuris). (Pugh et al., 2021)
Indeed, for us, it is not correct to write “gastrointestinal strongyle eggs”.
Nevertheless, in order to adjust as demand while being more specific, we changed for “gastrointestinal strongyle ‘type’ eggs” (with a quick explanation in the beginning of the Results).”
Line 230: replace “Parasites Load Response to Treatment” with “Quantitative Parasitological Analysis”
Done, Line 215. Pr Losson also suggested “Effect of G. ulmifolia supplementation on FEC and FOC”
Lines 232-236: please replace
“1. Gastrointestinal strongyle ‘type’ eggs: term which include GI strongyles with morphological 232 similarites, such as, for the most common genus, Haemonchus, Ostertagia, Trichostrongylus, and Cooperia [6].
- Strongyloides papillosus eggs
- Eimeria ookysts”
with
“1. Gastrointestinal strongyle eggs
- Strongyloides papillosus eggs
- Eimeria spp. oocysts”
Done for Eimeria. However, as explained previously the explanation of the “GI strongyle ‘type’ eggs” terminology is adequate for us, we think it is important to keep it.
Line 237: delete this sentence
Done
Line 240: please replace “gastrointestinal strongyle ‘type’ eggs load showed” with “the number of gastrointestinal strongyle EPG showed”
Done. Pr Losson suggested “gastrointestinal strongyle eggs counts showed”
Line 242: please replace “Eimeria spp. oocysts load also showed” with “the number of Eimeria spp. OPG also showed”
Done
Line 244: please replace “Finally, the load of Strongyloides papillosus eggs” with “Finally, the number of S. papillosus EPG”
Line 226, We changed for the suggestion of Pr Losson “eggs counts of S. papillosus”, in order not to repeat the same formulation as for the GI strongyle Line (Line 222).
Line 247: please replace “Figure 4. Evolution of parasitic load of gastrointestinal strongyle ‘type’ eggs (EPG) of sheep fed” with “Figure 4. Gastrointestinal strongyle eggs per gram of faeces (EPG) of sheep fed”
Done, we kept « Evolution of » in order to still have the time notion.
Pr. Losson proposal: “Evolution of gastrointestinal strongyle faecal egg counts (EPG) of sheep fed hay of C. nlemfuensis alone (G0) or foliage of G. ulmifolia and hay of C. nlemfuensis (G1).”
Lines 250-251: please replace “Figure 5. Evolution of parasitic load of Eimeria spp. oocysts (OPG) of sheep fed on C. nlemfuensis hay alone (G0) or G. ulmifolia foliage and C. nlemfuensis hay (G1).” with “Figure 5. Eimeria spp. oocysts per gram of faeces (OPG) of sheep fed on C. nlemfuensis hay alone (G0) or G. ulmifolia foliage and C. nlemfuensis hay (G1).”
Done, ditto.
Pr. Losson proposal: “Evolution of Eimeria spp. faecal oocysts counts (OPG) of sheep fed hay of C. nlemfuensis alone (G0) or foliage of G. ulmifolia and hay of C. nlemfuensis (G1).”
Lines 253-254: please replace “Figure 6. Parasitic load of Strongyloides papillosus eggs (EPG) of sheep fed on C. nlemfuensis hay alone (G0) or G. ulmifolia foliage and C. nlemfuensis hay (G1).” with “Figure 6. Strongyloides papillosus eggs per gram of faeces (EPG) of sheep fed on C. nlemfuensis hay alone (G0) or G. ulmifolia foliage and C. nlemfuensis hay (G1).”
Done, ditto.
Pr. Losson proposal: “Evolution of Strongyloides papillosus faecal egg counts (EPG) of sheep fed hay of C. nlemfuensis alone (G0) or foliage of G. ulmifolia and hay of C. nlemfuensis (G1).”
Discussion
Line 288: please replace “4.2. Feed Intake, Animal Performance, and Parasitic Load” with “4.2. Feed Intake, Animal Performance, and Gastrointestinal Parasites”
Done
Pr. Losson proposal: “Feed Intake, Animal Performance, and Egg/Oocyst shedding”
Line 297: please replace “parasitic load of nematodes” with “the EPG number of gastrointestinal strongyle eggs”
Modified by Pr. Losson proposal: “the GI strongyle egg shedding”
Line 298: replace “reaching significant parasitism at >1000 EPG” with “reaching a number >1000 EPG, indicating a possible high gastrointestinal strongyle load”
Modified by Pr. Losson proposal: “reaching a level (>1000 EPG) which is considered to be associated with heavy parasite burden and significant effects on health and productivity in sheep”
Line 312: please replace “parasitological load” with “parasite infections”
Done with adjustment. In this sentence we really referred to the number of parasites in the GI tract which is important for the lamb (in contrary to the adult due to the development of the immunity which regulate the number, as described just after in the text). However, infections are indeed also greater in lamb. Thus we write both.
The load of parasites is evaluated mainly by counting the number of parasites in the gastrointestinal tract. The number of nematode eggs or of coccidian oocysts give only an estimation of the load of these parasites. Therefore, replace “The GI parasites’ load was estimated by coprological analyses” with “Quantitative coprological analysis was performed for the evaluation of gastrointestinal parasites”.
Line 308: please replace “worm number” with “nematode (or helminth) number”
Done
Line 317: please replace “presently” with “of this study”
Done
Line 327: please replace “various worms” with “various nematode (or helminth) species
Done
Lines 334-336: please delete the sentence “In our experiment, sheep examined were infected principally by gastrointestinal strongyle ‘type’ species. This terminology, used with the McMaster technic results, include not only species localized in the abomasum, but mostly [7].” with “In our experiment, sheep examined were positive for gastrointestinal strongyle eggs at faecal parasitological analysis. However, the identification of these nematodes at the genus/species level is not possible on the basis of morphological features of their eggs [7].”
Done
Line 339: please replace “cattle (???)” with “sheep” and “worms” with “parasites”
Done
Lines 347-348: please replace “Considering the antiparasitological and nutritional benefits of CT present in G. ulmifolia foliage no apparent effect was highlighted.” with “Despite the antiparasitic and nutritional benefits reported for CT [insert references], no apparent effect was highlighted for CT contained in G. ulmifolia foliage tested in this study.”
Done
References
Reference n. 6 should be replaced with a more appropriate one (published by parasitologists in an international journal)
References n. 17 and 53 are the same
Sorry. Done.
In the reference n. 51 “Acacia Mearnsii” should be written in italics and the first letter of the name of the species should be in lowercase.
Thank you. Done.
